# To Prune, or Not to Prune: Exploring the Efficacy of Pruning for Model Compression

## Abstract

Model pruning seeks to induce sparsity in a deep neural network's various connection matrices, thereby reducing the number of nonzero-valued parameters in the model. Recent reports (Han et al., 2015a; Narang et al., 2017) prune deep networks at the cost of only a marginal loss in accuracy and achieve a sizable reduction in model size. This hints at the possibility that the baseline models in these experiments are perhaps severely over-parameterized at the outset and a viable alternative for model compression might be to simply reduce the number of hidden units while maintaining the model's dense connection structure, exposing a similar trade-off in model size and accuracy. We investigate these two distinct paths for model compression within the context of energy-efficient inference in resource-constrained environments and propose a new gradual pruning technique that is simple and straightforward to apply across a variety of models/datasets with minimal tuning and can be seamlessly incorporated within the training process. We compare the accuracy of large, but pruned models (*large-sparse*) and their smaller, but dense (*small-dense*) counterparts with identical memory footprint. Across a broad range of neural network architectures (deep CNNs, stacked LSTM, and seq2seq LSTM models), we find *large-sparse* models to consistently outperform *small-dense* models and achieve up to 10x reduction in number of non-zero parameters with minimal loss in accuracy.

## 1 Introduction

Over the past few years, deep neural networks have achieved state-of-the-art performance on several challenging tasks in the domains of computer vision, speech recognition, and natural language processing. Driven by increasing amounts of data and computational power, deep learning models have become bigger and deeper to better learn from data. While these models are typically deployed in a datacenter back-end, preserving user privacy and reducing user-perceived query times mandate the migration of the intelligence offered by these deep neural networks towards edge computing devices. Deploying large, accurate deep learning models to resource-constrained computing environments such as mobile phones, smart cameras etc. for on-device inference poses a few key challenges. Firstly, state-of-the-art deep learning models routinely have millions of parameters requiring ~MBs of storage, whereas on-device memory is limited. Furthermore, it is not uncommon for even a single model inference to invoke ~billions of memory accesses and arithmetic operations, all of which consume power and dissipate heat which may drain the limited battery capacity and/or test the device's thermal limits.

Confronting these challenges, a growing body of work has emerged that intends to discover methods for compressing neural network models while limiting any potential loss in model quality. Latency-sensitive workloads relying on energy-efficient on-device neural network inference are often memory bandwidth-bound, and model compression offers the two-fold benefit of reducing the total number of energy-intensive memory accesses as well as improving the inference time due to an effectively higher memory bandwidth for fetching compressed model parameters. Within the realm of model compression techniques, pruning away (forcing to zero) the less salient connections (parameters) in the neural network has been shown to reduce the number of nonzero parameters in the model with little to no loss in the final model quality. Model pruning enables trading off a small degradation in model quality for a reduction in model size, potentially reaping improvements in inference time and energy-efficiency. The resulting pruned model typically has sparse connection matrices,

so efficient inference using these sparse models requires purpose-built hardware capable of loading sparse matrices and/or performing sparse matrix-vector operations (Zhang et al., 2016; Han et al., 2016; Parashar et al., 2017). Also, representing sparse matrices carries with it an additional storage overhead increasing the model's net memory footprint which must also be taken into consideration.

In this work, we perform a closer examination of the effectiveness of model pruning as a means for model compression. *From the perspective of on-device neural network inference, given a bound on the model's memory footprint, how can we arrive at the most accurate model?* We aim to answer this question by comparing the quality of the models obtained through two distinct methods: (1) training a large model, but pruned to obtain a sparse model with a small number of nonzero parameters (*large-sparse*); and (2) training a *small-dense* model with size comparable to the *large-sparse* model. Both of these methods expose a model accuracy and size tradeoff, but differ remarkably in terms of their implications on the design of the underlying hardware architecture. For this comparative study, we pick models across a diverse set of application domains: InceptionV3 (Szegedy et al., 2016) and MobileNets (Howard et al., 2017) for image recognitions tasks, stacked LSTMs for language modeling, and seq2seq models used in Google's Neural Machine Translation (Wu et al., 2016) system. In the process of this investigation, we also develop a simple gradual pruning approach that requires minimal tuning and can be seamlessly incorporated within the training process and demonstrate its applicability and performance on an assortment of neural network architectures.

## 2    RELATED WORK

Early works in the 1990s (LeCun et al., 1990; Hassibi et al., 1993) performed pruning using a second-order Taylor approximation of the increase in the loss function of the network when a weight is set to zero. In Optimal Brain Damage (LeCun et al., 1990), the saliency for each weight was computed using a diagonal Hessian approximation, and the low-saliency weights were pruned from the network and the network was retrained. In Optimal Brain Surgeon (Hassibi et al., 1993), the saliency for each weight was computed using the inverse Hessian matrix, and the low-saliency weights were pruned and all other weights in the network were updated using the Hessian matrix.

More recently, magnitude-based weight pruning methods have become popular techniques for network pruning (Han et al., 2015b;a; See et al., 2016; Narang et al., 2017). Magnitude-based weight pruning techniques are computationally efficient, scaling to large networks and datasets. Our automated gradual pruning algorithm prunes the smallest magnitude weights to achieve a preset level of network sparsity. In contrast with the works listed above, our paper focuses on comparing the model accuracy and size tradeoff of *large-sparse* versus *small-dense* models.

A work similar to ours is the work by Narang et al. (2017) on pruning a RNN and GRU model for speech recognition and showing that a sparse RNN that was pruned outperformed a dense RNN trained normally of comparable size. While they provide one data point comparing the performance of a sparse vs dense model, our work does an extensive comparison of sparse vs dense models across a wide range of models in different domains (vision and NLP). Narang et al. also introduce a gradual pruning scheme based on pruning all the weights in a layer less than some threshold (manually chosen) which is linear with some slope in phase 1 and linear with some slope in phase 2 followed by normal training. Compared to their approach, we do not have two phases and do not have to choose two slopes, and we do not need to choose weight thresholds for each layer (we rely on a sparsity schedule which determines the weight thresholds). Thus, our technique is simpler, doesn't require much hyperparameter tuning, and is shown to perform well across different models.

Within the context of reducing model size by removing redundant connections, several recent works (Anwar et al., 2015; Lebedev & Lempitsky, 2015; Li et al., 2016; Changpinyo et al., 2017) propose techniques to prune and induce sparsity in a structured way, motivated primarily by the desire to speedup computations on existing hardware architectures optimized for dense linear algebra. Such techniques perform coarse-grain pruning and depend critically on the structure of the convolutional layers, and may not be directly extensible to other neural network architectures that lack such structural properties (LSTMs for instance). On the contrary, our method does not make any assumptions about the structure of the network or its constituent layers and is therefore more generally applicable.

While pruning focuses on reducing the number of non-zero parameters, in principle, model pruning can be used in conjunction with other techniques to further reduce model size. Quantization tech-

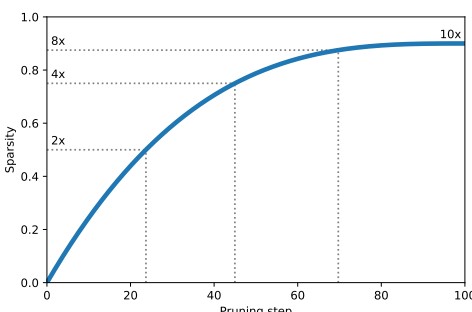

Table 1: Model size and accuracy tradeoff for sparse-InceptionV3

| Sparsity | NNZ params | Top-1 acc. | Top-5 acc. |
|---|---|---|---|
| 0% | 27.1M | 78.1% | 94.3% |
| 50% | 13.6M | 78.0% | 94.2% |
| 75% | 6.8M | 76.1% | 93.2% |
| 87.5% | 3.3M | 74.6% | 92.5% |

Figure 1: Sparsity function used for gradual pruning

niques aim to reduce the number of bits required to represent each parameter from 32-bit floats to 8 bits or fewer. Different quantization techniques such as fixed-point quantization (Vanhoucke et al., 2011) or vector quantization (Gong et al., 2014) achieve different compression ratios and accuracies but also require different software or hardware to support inference at runtime. Pruning can be combined with quantization to achieve maximal compression (Han et al., 2015a). In addition, an emerging area of research is low precision networks where the parameters and/or activations are quantized to 4 bits or fewer (Courbariaux et al., 2015; Lin et al., 2015; Hubara et al., 2016; Rastegari et al., 2016; Zhu et al., 2016). Besides quantization, other potentially complementary approaches to reducing model size include low-rank matrix factorization (Denil et al., 2013; Denton et al., 2014; Jaderberg et al., 2014; Lebedev et al., 2014) and group sparsity regularization to arrive at an optimal layer size (Alvarez & Salzmann, 2016).

## 3 METHODS

We extend the TensorFlow (Abadi et al., 2015) framework to prune the network's connections during training. For every layer chosen to be pruned, we add a binary mask variable which is of the same size and shape as the layer's weight tensor and determines which of the weights participate in the forward execution of the graph. We inject ops into the TensorFlow training graph to sort the weights in that layer by their absolute values and mask to zero the smallest magnitude weights until some desired sparsity level $s$ is reached. The back-propagated gradients flow through the binary masks, and the weights that were masked in the forward execution do not get updated in the back-propagation step. We introduce a new automated gradual pruning algorithm in which the sparsity is increased from an initial sparsity value $s_i$ (usually 0) to a final sparsity value $s_f$ over a span of $n$ pruning steps, starting at training step $t_0$ and with pruning frequency $\Delta t$:

$$s_t = s_f + (s_i - s_f) \left(1 - \frac{t - t_0}{n \Delta t}\right)^3 \quad \text{for} \quad t \in \{t_0, \ t_0 + \Delta t, \ ..., \ t_0 + n \Delta t\} \quad (1)$$

The binary weight masks are updated every $\Delta t$ steps as the network is trained to gradually increase the sparsity of the network while allowing the network training steps to recover from any pruning-induced loss in accuracy. In our experience, varying the pruning frequency $\Delta t$ between 100 and 1000 training steps had a negligible impact on the final model quality. Once the model achieves the target sparsity $s_f$, the weight masks are no longer updated. The intuition behind this sparsity function in equation (1) is to prune the network rapidly in the initial phase when the redundant connections are abundant and gradually reduce the number of weights being pruned each time as there are fewer and fewer weights remaining in the network, as illustrated in Figure 1. In the experimental results presented in this paper, pruning is initiated after the model has been trained for a few epochs or from a pre-trained model. This determines the value for the hyperparameter $t_0$. A suitable choice for $n$ is largely dependent on the learning rate schedule. Stochastic gradient descent (and its many variants) typically decay the learning rate during training, and we have observed that pruning in the presence of an exceedingly small learning rate makes it difficult for the subsequent training steps to recover

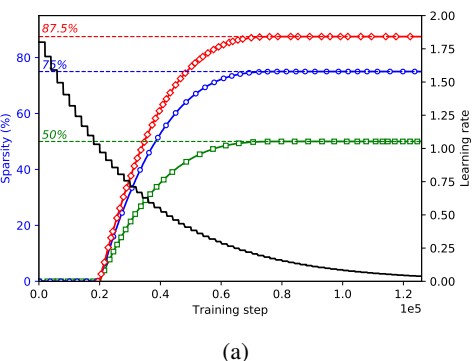 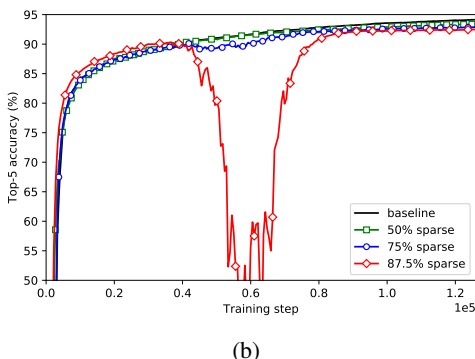

(a) (b)

Figure 2: (a) The gradual sparsity function and exponentially decaying learning rate used for training sparse-InceptionV3 models. (b) Evolution of the model's accuracy during the training process

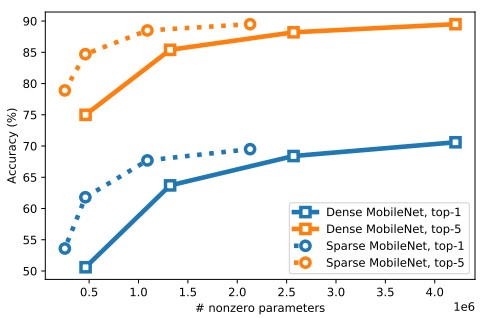 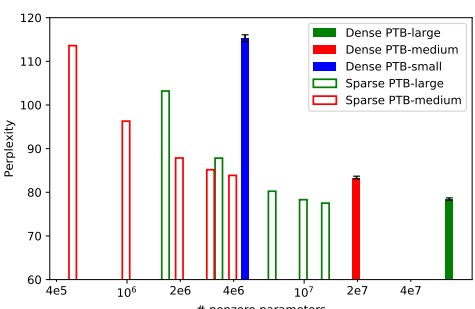

Figure 3: MobileNet sparse vs dense results    Figure 4: PTB sparse vs dense results

from the loss in accuracy caused by forcing the weights to zero. At the same time, pruning with too high of a learning rate may mean pruning weights when the weights have not yet converged to a good solution, so the pruning schedule should be chosen closely with the learning rate schedule.

Figure 2a shows the learning rate and the pruning schedule used for training sparse-InceptionV3 (Szegedy et al., 2016) models. All the convolutional layers in this model are pruned using the same sparsity function, and pruning occurs in the regime where the learning rate is still reasonably high to allow the network to heal from the pruning-induced damage. Figure 2b offers more insight into how this pruning scheme interacts with the training procedure. For the 87.5% sparse model, with the gradual increase in sparsity, there comes a point when the model suffers a near-catastrophic degradation, but recovers nearly just as quickly with continued training. This behavior is more pronounced in the models trained to have higher sparsity. Table 1 compares the performance of sparse-InceptionV3 models pruned to varying extents. As expected, there is a gradual degradation in the model quality as the sparsity increases. However, a 50% sparse model performs just as well as the baseline (0% sparsity), and there is only a 2% decrease in top-5 classification accuracy for the 87.5% sparse model which offers an 8x reduction in number of nonzero (NNZ) model parameters. Also note that since the weights are initialized randomly, the sparsity in the weight tensors does not exhibit any specific structure. Furthermore, the pruning method described here does not depend on any specific property of the network or the constituent layers, and can be extended directly to a wide-range of neural network architectures.

## 4   COMPARING *large-sparse* AND *small-dense* MODELS

### 4.1   MOBILENETS

MobileNets are a class of efficient convolutional neural networks designed specifically for mobile vision applications (Howard et al., 2017). Instead of using standard convolutions, MobileNets are

Table 2: MobileNet sparse vs dense results

| Width | Sparsity | NNZ params | Top-1 acc. | Top-5 acc. |
|---|---|---|---|---|
| 0.25 | 0% | 0.46M | 50.6% | 75.0% |
| 0.5 | 0% | 1.32M | 63.7% | 85.4% |
| 0.75 | 0% | 2.57M | 68.4% | 88.2% |
| 1.0 | 0% | 4.21M | 70.6% | 89.5% |
| | 50% | 2.13M | 69.5% | 89.5% |
| | 75% | 1.09M | 67.7% | 88.5% |
| | 90% | 0.46M | 61.8% | 84.7% |
| | 95% | 0.25M | 53.6% | 78.9% |

Table 3: PTB sparse vs dense results

| Model | Sparsity | NNZ params | Per-plexity |
|---|---|---|---|
| Small | 0% | 4.6M | 115.30 |
| Medium | 0% | 19.8M | 83.37 |
| | 80% | 4.0M | 83.87 |
| | 85% | 3.0M | 85.17 |
| | 90% | 2.0M | 87.86 |
| | 95% | 1.0M | 96.30 |
| | 97.5% | 0.5M | 113.6 |
| Large | 0% | 66M | 78.45 |
| | 80% | 13.2M | 77.52 |
| | 85% | 9.9M | 78.31 |
| | 90% | 6.6M | 80.24 |
| | 95% | 3.3M | 87.83 |
| | 97.5% | 1.7M | 103.20 |

based on a form of factorized convolutions called depthwise separable convolution. Depthwise separable convolutions consist of a depthwise convolution followed by a 1x1 convolution called a pointwise convolution. This factorization significantly reduces the number of parameters in the model by filtering and combining input channels in two separate steps instead of together as in the standard convolution. The MobileNet architecture consists of one standard convolution layer acting on the input image, a stack of depthwise separable convolutions, and finally averaging pooling and fully connected layers. For the dense baseline model with width multiplier 1.0, there are a total of 4.21M parameters, 99% of which are in the 1x1 pointwise convolution layers (74.6%) and fully connected layers (24.3%). We do not prune the parameters in the one standard convolution layer and in the depthwise convolution layers since there are very few parameters in those layers (1.1%).

The width multiplier is a parameter of the MobileNet network that allows trading off the accuracy of the model with the number of parameters and computational cost. The width multiplier of the baseline model is 1.0. For a given width multiplier $\alpha \in (0, 1]$, the number of input channels and the number of output channels in each layer is scaled by $\alpha$ relative to the baseline 1.0 model. We compare the performance of dense MobileNets trained with width multipliers 0.75, 0.5, and 0.25 with the performance of sparse MobileNets pruned from dense 1.0 MobileNet in Figure 3 and Table 2 on the ImageNet dataset. We see that for a given number of non-zero parameters, sparse MobileNets are able to outperform dense MobileNets. For example, the 75% sparse model (which has 1.09 million parameters and a top-1 accuracy of 67.7%) outperforms the dense 0.5 MobileNet (which has 1.32 million parameters and a top-1 accuracy of 63.7%) by 4% in top-1 accuracy while being smaller. Similarly, the 90% sparse model (which has 0.46 million parameters and a top-1 accuracy of 61.8%) outperforms the dense 0.25 MobileNet (which has 0.46 million parameters and a top-1 accuracy of 50.6%) by 10.2% in top-1 accuracy while having the same number of non-zero parameters.

Overall, pruning is a promising approach for model compression even for an architecture that was designed to be compact and efficient by using depthwise separable convolutions instead of standard convolutions as a factorization-like technique to reduce the number of parameters. The sparsity parameter is shown to be an effective way to trade off the accuracy of a model with its memory usage and compares favorably with the width multiplier in MobileNet. Training a sparse MobileNet using our gradual pruning algorithm is also easy. For pruning a dense MobileNet, we used the same learning rate schedule as for training a dense MobileNet but with an initial learning rate 10 times smaller than for training a dense MobileNet, and all other hyperparameters were kept the same.

## 4.2 PENN TREE BANK (PTB) LANGUAGE MODEL

We train an LSTM language model on the Penn Tree Bank dataset using the models and training procedure described in Zaremba et al. (2014). At each time step, the LSTM language model outputs the probability of the next word in the sentence given the history of previous words. The loss func-

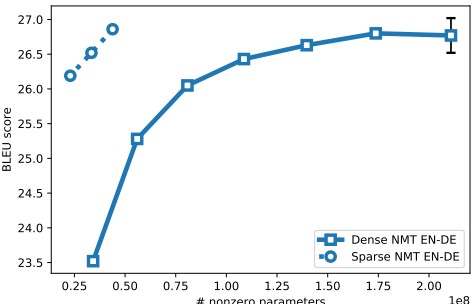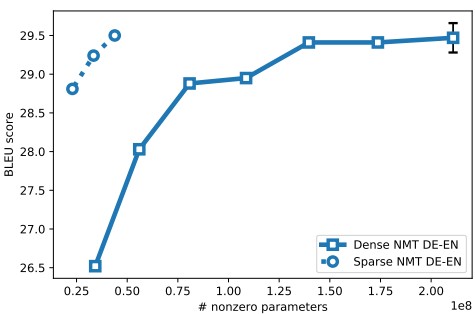

Figure 5: Comparison of sparse vs dense NMT models for English to German (EN-DE) and German to English (DE-EN) translation

tion is the average negative log probability of the target words, and the perplexity is the exponential of the loss function. The language model is composed of an embedding layer, 2 LSTM layers, and a softmax layer. The vocabulary size is 10,000, and the LSTM hidden layer size is 200 for the small model, 650 for the medium model, and 1,500 for the large model. In the case of the large model, there are 15M parameters in the embedding layer, 18M parameters in each of the two LSTM layers, and 15M parameters in the softmax layer for a total of 66M parameters. Different hyperparameters are used to train the different-sized models. When pruning a model of a certain size, we use the same hyperparameters that were used for training the dense model of that size. We compare the performance of the dense models with sparse models pruned from medium and large to 80%, 85%, 90%, 95%, and 97.5% sparsity in Figure 4 and Table 3. In this case, we see that sparse models are able to outperform dense models which have significantly more parameters (note the log scale for the number of parameters). The 90% sparse large model (which has 6.6 million parameters and a perplexity of 80.24) is able to outperform the dense medium model (which has 19.8 million parameters and a perplexity of 83.37), a model which has 3 times more parameters. Compared with MobileNet, pruning PTB model likely gives better results because the PTB model is larger with significantly more parameters. Our results show that pruning works very well not only on the dense LSTM weights and dense softmax layer but also the dense embedding matrix. This suggests that during the optimization procedure the neural network can find a good sparse embedding for the words in the vocabulary that works well together with the sparse connectivity structure of the LSTM weights and softmax layer.

From Figure 4 and Table 3, we also see that the 85% sparse medium model (which has 3 million parameters and a perplexity of 85.17) outperforms the 95% sparse large model (which has 3.3 million parameters and a perplexity of 87.83). The accuracy of the 95% sparse large model is comparable to the accuracy of the 90% sparse medium model (which has 2 million parameters and a perplexity of 87.86). Together, these results suggest that there is an optimal compression range when pruning. In the case of PTB, pruning to 95% sparsity for a compression ratio of 20x significantly degrades the performance of the sparse model compared to pruning to 90% sparsity for a compression ratio of 10x, as seen in Figure 4 from the curve of perplexity vs. number of parameters traced by either of the sparse models. These results suggest that in order to get the best-performing sparse model of a certain size, we should train a dense model that is 5x-10x larger and then prune to the desired number of parameters rather than taking the largest and best-performing dense model and pruning this model by 20x or more to the desired number of parameters, assuming that the difference in performance of the two dense baseline models is not that large. We note that it may be possible to obtain slightly better results for pruning to 95% sparsity or higher with more hyperparameter tuning, and the results we obtained for pruning a model of a certain size were from using exactly the same hyperparameter configuration as for training the dense model of that size.

## 4.3    GOOGLE NEURAL MACHINE TRANSLATION

The Google Neural Machine Translation (NMT) architecture is a seq2seq model with attention (Wu et al., 2016). We use the open-source TensorFlow implementation available at Luong et al. (2017). The model is based on an encoder-decoder architecture. The encoder has an embedding layer which maps the source vocabulary of 36,548 words into a $k$-dimensional space, 1 bidirectional LSTM

Table 4: NMT sparse vs dense results

| # units | Sparsity | NNZ params | EN-DE BLEU score | DE-EN BLEU score |
|---------|----------|------------|------------------|------------------|
| 256 | 0% | 34M | 23.52 | 26.52 |
| 512 | 0% | 81M | 26.05 | 28.88 |
| 768 | 0% | 140M | 26.63 | 29.41 |
| 1024 | 0% | 211M | 26.77 | 29.47 |
|  | 80% | 44M | 26.86 | 29.50 |
|  | 85% | 33M | 26.52 | 29.24 |
|  | 90% | 23M | 26.19 | 28.81 |

layer, and 3 standard LSTM layers. The decoder has an embedding layer which maps the target vocabulary of 36,548 words into a $k$-dimensional space, 4 LSTM layers with attention, and finally a softmax layer. For the dense baseline model with number of units $k = 1024$, there are 37.4M parameters in each of the encoder embedding, decoder embedding, and softmax layers and 98.6M parameters in all of the LSTM layers for a total of 211M parameters. We apply pruning to all of the LSTM layers, embedding layers, and softmax layers, but we do not prune the attention parameters of which there are relatively few. The other dense models were obtained by varying the number of units $k$. We use the WMT16 German and English dataset with news-test2013 as the dev set and news-test2015 as the test set. The BLEU score is reported as a measure of the translation quality. The learning rate schedule used for training the dense models is 170K iterations with initial learning rate 1.0 and 170K iterations with learning rate decay of 0.5 every 17K iterations. For pruning a dense model, the learning rate schedule we use is 70K iterations with initial learning rate 0.5 and 170K iterations with learning rate decay of 0.5 every 17K iterations, and all other hyperparameters were kept the same.

Since we noticed that the NMT training procedure had high variance, we tested several pruning schemes applied to NMT. Our standard implementation of gradual pruning increases the sparsity of every layer to the same sparsity level at each pruning step. We tested a variant which we call "layerwise constant" sparsity: instead of simultaneously increasing the sparsity of all layers to some sparsity level at each pruning step, we subdivide the pruning interval and increase the sparsity of one layer at a time to that sparsity level. This potentially has the effect of reducing the impact of pruning and allowing the network to recover better with training. Finally, we compared with "global" pruning: we prune the smallest magnitude weights across the entire network, regardless of which layer they are in. Global pruning produces a different sparsity level for each layer and was shown to perform well on NMT in the work of See et al. (2016). Overall, the layerwise constant pruning scheme performed best on average, so we report the results with the layerwise constant pruning scheme in Figure 5 and Table 4. We note that there is high variance in the results due to the stochasticity of the training process, as illustrated by the error bar in Figure 5 which is the standard deviation of the BLEU score of 10 randomly initialized and independently trained NMT models.

The results in Table 4 show that for 80% sparsity (5x compression), the pruned model actually achieves a slightly higher BLEU score than the baseline model (though we note the error bar). For 85% sparsity, the BLEU score drops by around 0.25, and for 90% sparsity, the BLEU score drops by around 0.6. When we compare the performance of dense and sparse models in Figure 5 and Table 4, we again see that sparse models outperform even larger-sized dense models. The BLEU score of the dense model falls off quickly after 2x reduction in model size while the BLEU score of the sparse model starts to fall off only after 5x reduction in NNZ parameters. For example, the 90% sparse 1024-unit model is comparable to or outperforms the dense 512-unit model (26.19 vs 26.05 for EN-DE and 28.81 vs 28.88 for DE-EN) despite having 3.5x fewer NNZ params (23M vs 81M).

## 5 DISCUSSION

The net memory footprint of a sparse model includes the storage for the nonzero parameters and any auxiliary data structures needed for indexing these elements. Pruning models helps reduce the number of nonzero-valued connections in the network; however the overhead in sparse matrix storage

Table 5: Storage overheads associated with bit-mask and CSR(C) sparse matrix representations for sparse-MobileNets

| Sparsity | NNZ params | Bit-mask (MB) | CSR(C) (MB) |
|----------|-----------|---------------|-------------|
| 0%  | 4.21M | N/A  | N/A  |
| 50% | 2.13M | 0.52 | 1.06 |
| 75% | 1.09M | 0.52 | 0.54 |
| 90% | 0.46M | 0.52 | 0.23 |
| 95% | 0.25M | 0.52 | 0.13 |

Table 6: Comparison of the performance of *small-dense* and *large-sparse* models. Model size calculations include overhead for sparse matrix storage and assumes 32-bit (4 bytes) per nonzero element.

| Model | Small-dense | | Large-sparse | |
|-------|-------------|--------------|--------------|--------------|
| | Model size (MB) | Accuracy (%) | Model size (MB) | Accuracy (%) |
| | 10.28 | 68.4 | 9.04 | **69.5** |
| MobileNet | 5.28 | 63.7 | 4.88 | **67.7** |
| | 1.84 | 50.6 | 2.07 | **61.8** |
| | | | 1.13 | **53.6** |

inevitably diminishes the achievable compression ratio. The bit-mask sparse matrix representation requires 1 bit per matrix element indicating whether the element is nonzero, and a vector containing all the nonzero matrix elements. This representation incurs a constant overhead regardless of the model sparsity. In the compressed sparse row (column) storage (CSR(C)) adopted in Parashar et al. (2017), each nonzero parameter in the sparse matrix is associated with a count (usually stored as a 4 or 5 bit integer) of the number of zeros preceding it. The overhead in this case is proportional to the NNZ in the model. Table 5 compares these two representations for sparse-MobileNets. The CSR(C) representation can enable higher compression ratio for networks with high sparsity. Note, however, that the bit-mask representation offers marginally lower overhead at smaller sparsity levels.

In spite of this overhead, *large-sparse* models appear to achieve higher accuracy than *small-dense* models with comparable memory footprint. For instance, MobileNet with width multiplier 1 and sparsity 50% has similar footprint as MobileNet with width multiplier 0.75, but obtains higher accuracy. Table 6 further highlights the trade-off between model size and accuracy for dense and sparse models. The performance gap between *large-sparse* and *small-dense* models widens for larger models such as as the PTB language models and NMT (see Table 3 and Table 4). It is worth noting that the results presented in this work were obtained by training neural networks using 32-bit floating point representation. For neural networks trained to perform inference using reduced precision (8-bit integer, for instance) arithmetic, the memory overhead of sparse matrix storage represents a bigger fraction of the total memory footprint. Quantization of the parameters to a reduced precision number representation is also an effective method for model compression, and the interplay between model quantization and pruning and their collective impact on model accuracy merits a closer examination. We defer that investigation to a future extension to this work.

## 6 CONCLUSION

This work sheds light on the model size and accuracy trade-off encountered in pruned deep neural networks. We demonstrate that *large-sparse* models outperform comparably-sized *small-dense* models across a diverse set of neural network architectures. We also present a gradual pruning technique that can be applied with ease across these different architectures. We believe these results will encourage the adoption of model pruning as a tool for compressing neural networks for deployment in resource-constrained environments. At the same time, we hold the opinion that our results will provide further impetus to the hardware architecture community to customize the next generation of deep learning accelerator architectures to efficiently handle sparse matrix storage and computations.

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
