# OpenReview forum: "To Prune, or Not to Prune: Exploring the Efficacy of Pruning for Model Compression"
_ICLR.cc/2018/Conference — Invite to Workshop Track_

### Official Review · AnonReviewer3 · 2017-11-27
**pruning efficacy in deep learning**

**Rating:** 5
**Confidence:** 4

**Review:**

This paper presents a comparison of model sizes and accuracy variation for pruned version of over-parameterized deep networks and smaller but dense models of the same size. It also presents an algorithm for gradual pruning of small magnitude weight to achieve a pre-determined level of sparsity. The paper demonstrates that pruning of large over-parameterized models leads to better classification compared to smaller dense models of relatively same size. This pruning technique is demonstrated as a modification to TensorFlow on MobileNet, LSTM for PTB dataset and NMT for seq2seq modeling.

The paper seems mainly a comparison of impact of pruning a large model for various tasks. The novelty in the work seems quite limited mainly in terms of tensorflow implementation of the network pruning using a binary mask. The weights which are masked in the forward pass don't get updated in the backward pass. The fact that most deep networks are inherently over-parametrized seems to be known for quite sometime.

The experiments are missing comparison with the threshold based pruning proposed by Han etal. to ascertain if the gradual method is indeed better. A computational complexity comparison is also important if the proposed pruning method is indeed effective. In Section 1, the paper claims to arrive at "the most accurate model". However, the validation of the claim is mostly empirical and shows that there lies a range of values for increase in sparsity and decrease in prediction accuracy is better compared to other values.

Overall, the paper seems to perform experimental validation of some of the known beliefs in deep learning. The novelty in terms of ideas and insights seems quite limited.

---

### Official Review · AnonReviewer1 · 2017-11-27
**Informative experiments, unsurprising results.**

**Rating:** 5
**Confidence:** 4

**Review:**

Summary:
This paper presents a thorough examination of the effects of pruning on model performance.  Importantly, they compare the performance of "large-sparse" models (large models that underwent pruning in order to reduce memory footprint of model) and "small-dense" models, showing that "large-sparse" models typically perform better than the "small-dense" models of comparable size (in terms of number of non-zero parameters, and/or memory footprint).  They present results across a number of domains (computer vision, language modelling, and neural machine translation) and model types (CNNs, LSTMs).  They also propose a way of performing pruning with a pre-defined sparsity schedule, simplifying the pruning process in a way which works across domains.  They are able to show convincingly that pruning is an effective way of trading off accuracy for model size (more effective than simply reducing the size of model architecture), although there does come a point where too much sparsity degrades the model performance considerably; this suggests that pruning a medium size model to 80%-90% sparsity is likely better than pruning a larger model to >= 95% sparsity.

Review:
Quality: The quality of the work is high --- the experiments are extensive and thorough.  I would have liked to see "small-dense" vs. "large-sparse" comparisons on Inception (only large-sparse results are reported).

Clarity: The paper is clearly written, though there is room for improvement. For example, many of the results are presented in a redundant manner (in both tables and figures, where the table and figure are often not next to each other in the document).  Also, it is not clear in several cases exactly which training/heldout/test sets are used, and on which partition of the data the accuracies/BLEU scores/perplexities presented correspond to. A small section (before "Methods") describing the datasets/features in detail would be helpful.  Also, it would have probably been nice to explain all of the tasks and datasets early on, and then present all the results at once (NIT: include the plots in paper, and move the tables to an appendix).

Originality: Although the experiments are informative, the work as a whole is not very original.  The method proposed of using a sparsity schedule to perform pruning is simple and effective, but is a rather incremental contribution. The primary contribution of this paper is its experiments, which for the most part compare known methods.

Significance: The paper makes a nice contribution, though it is not particularly significant or surprising.  The primary observations are:
(1) large-sparse is typically better than small-dense, for a fixed number of non-zero parameters and/or memory footprint.
(2) There is a point at which increasing the sparsity percentage severely degrades the performance of the model, which suggests that there is a "sweet-spot" when it comes to choosing the model architecture and sparsity percentage which give the best performance (for a fixed memory footprint).

Result #1 is not very surprising, given that Han et al (2016) were able to show significant compression without loss in accuracy; thus, because one would expect a smaller dense model to perform worse than the large dense model, it would also perform worse than the large sparse model.
Result #2 had already been seen in Han et al (2016) (for example, in Figure 6).

Pros:
- Very thorough experiments across a number of domains

Cons:
- Methodological contributions are minor.
- Results are not surprising, and are in line with previous papers.

---

### Official Review · AnonReviewer2 · 2017-11-28
**This paper analyzes the effectiveness of model pruning for deployment in resource constrained environments. The contribution is marginal but interesting as a summary**

**Rating:** 5
**Confidence:** 5

**Review:**

This paper analyzes the effectiveness of model pruning for deployment in resource constrained environments. The contribution is marginal but interesting as a summary


This paper analyzes the effectiveness of model pruning for deployment in resource constrained environments. Contrary to other approaches, this paper assumes there is a computational budget to be meet and the pruning approach should result in a model that fits within that budget.

According to the paper there is a contribution of a pruning scheme. To the best of my understanding, the proposal / contribution is minimal or not clearly detailed. My understanding is the approach is equivalent to a L1 pruning where the threshold for pruning is updated over time / training process rather than pushing weights down towards zero (as it is usually done).
Then, there is a schedule for minimizing the impact of modifying the weights although this has been discussed in related works (see Alvarez and Salzmann 2016).


Given this setup, the paper present a number of comparisons and experimental validations.

There are several steps that are not clear to me.

1) how does this compare to the low-rank or group sparsity approaches referred in the related work section?
2) The key here is modifying the thresholds as the training progresses up to a certain point which seems to me quite equivalent to L1 pruning where the regularization term is also affected by the learning rate (therefore having less influence as the training progresses). In this paper though there are heuristics to stop pruning when certain constraints are met. Which is interesting (as pruning will affect the quality and capacity of the network) but also applicable to other methods. Also, as suggested in related works, the pruning becomes negligible after certain number of epochs (therefore there is no real need to stop the process). Any discussion here would be interesting.

3) For me, it is interesting the fact that pruning in an initial stage is too aggressive. However, it also limits the capacity of the network by pruning too much at the begining. I think there are contrary messages in the paper that would be nice to clarify: pruning rapidly at the beginning where redundant connections are abundant and then, there is the need to have a large learning rate to recover from the pruning.

4) I missed a discussion on the Inception model in the experimental settings.

5) If this is based on masks for pruning and performing sparse operations I wonder how does this benefit at inference time since many operations will be faster in a dense matrix multiplication manner. That is why I think would be interesting to do at group level as proposed in some related methods.

6) Tables showing comparisons are not complete. I do not understand why measuring the non-zero parameters if, in the baseline, there is no analysis on how many of these parameters can be actually set to 0 by pruning as a postprocessing step. Please, add explanations on why / how non-zeros are measured in the baseline.

7) More importantly, I think the comparison sparsity vs width is not fair. This is comparing the training process of a model with limited capacity vs a model where the capacity is progressively limited (the pruned). Training regimes should be detailed and properly analyzed for this to be fair. Nevertheless, results are consistent with other approaches listed in the state of the art (pruning while training is a good thing).

---

> ### Author Response · Authors · 2018-01-05
> **Response to Reviewer 2**
>
> Our proposed pruning method sets to zero the smallest magnitude weights in the weight tensor of each layer until a desired sparsity level s(t) is reached. By ensuring that 100*s(t)% of the weights are zero at time t, we avoid the need to tune per-layer weight thresholds or threshold hyperparameters. The gradual pruning is embedded during the training process of the respective network by pruning to sparsity s(t) once every Δt training steps (for all other iterations, the training step of the network is not changed). The sparsity function s(t) and the gradual pruning method are described in the Methods section.
>
> 1. Based on our literature review, we believe that magnitude-based weight pruning generally achieves better accuracy compared to low-rank and group sparsity approaches given the same sparsity constraints (though we did not test other methods ourselves).
>
> 2. See the first paragraph of our response. The key difference is that our pruning method is based on the sparsity function which directly controls the number of zeros in the weight tensors of each layer. We do not need to tune a L1 regularization coefficient or weight threshold hyperparameters.
>
> 3. We can begin pruning when the model is partially or fully trained. Pruning rapidly at the beginning means that we reduce the capacity of our network rapidly at the beginning. Since we have zeroed out a large number of weights in our network and perturbed our system by a large amount, we use a large learning rate to allow the network to recover from our large perturbation.
>
> 4. We did not compare small-dense and large-sparse Inception models so Inception is not in the experimental section, but we instead used the Inception model to demonstrate how our gradual pruning technique works in the Methods section.
>
> 5. We are interested in the potential of custom hardware architectures that support using sparse neural network models for on-device inference. Since on-device neural network inference is often memory bandwidth-bound, using sparse models can potentially lead to big speedups at inference time by reducing the number of parameters that are fetched from memory in each forward pass (we benefit even if the matrix multiplication is subsequently done using dense matrices if inference is memory-bound rather than compute-bound). Furthermore, by reducing the total number of energy-intensive memory accesses, we reduce the power consumption which is the more critical constraint for on-device neural network inference. The potential advantage of sparse models over group sparse models is that higher accuracy might be obtainable by using sparse models compared to group sparse models for the same memory footprint because group sparsity is a more restrictive condition than sparsity.
>
> 6. For the baseline (dense) model, the number of nonzero parameters is equal to the total number of parameters in the model. When using our proposed pruning method to train sparse models, we directly set many of the weights to zero, which makes the weight tensors sparse and reduces the number of nonzero parameters, without needing any postprocessing step.
>
> 7. In all of our experiments, the number of iterations for pruning a dense model (training a sparse model) is less than or equal to the number of iterations for originally training the respective dense model, so we have to tried to ensure a fair comparison between small-dense and large-sparse models. Furthermore, all hyperparameters, other than possibly the number of iterations and the learning rate schedule, are kept the same between the training of dense and sparse models.

---

### Public Comment · (anonymous) · 2017-11-01
**Supplementary comments on learning structured sparsity in DNNs**

Regarding " Such techniques perform coarse-grain pruning and depend critically on the structure of the convolutional layers, and may not be directly extensible to other neural network architectures that lack such structural properties (LSTMs for instance)",  the category of learning structured sparsity [1][2] in DNNs is a more general way than we thought. It is more challenging for DNNs with more sophisticated structures, but it might be possible to use it in LSTMs [3] and to even learn to reduce layers in ResNets [2], if we could figure out the kind of structures we want to learn.

[1] https://arxiv.org/abs/1608.08710
[2] http://papers.nips.cc/paper/6504-learning-structured-sparsity-in-deep-neural-networks.pdf
[3] https://arxiv.org/abs/1709.05027

---

### Author Response · Authors · 2018-01-05
**Response to reviewers**

We thank the reviewers for their time and feedback.

We want to emphasize what we view as the major contributions of our paper.

1. We demonstrate that for a constant model memory footprint, large-sparse models outperform small-dense models across several state-of-the-art neural network architectures. In hindsight, it may appear that this result is intuitive. However, our work provides extensive empirical proof of this property of deep neural networks across a diverse set of neural network architectures (as opposed to limiting the study to CNNs as in several of the prior works). To our knowledge, the fact that magnitude-based pruning can achieve high compression ratios with minimal loss in accuracy on state-of-the-art models like Inception, MobileNet, and Google NMT has not been previously shown in the literature. In the case of CNNs, Han et al. (2015) present pruning results on ImageNet using older CNNs such as AlexNet (42.8% top-1 error, 244MB) and VGGNet (31.5% top-1 error, 552 MB). In contrast, we present pruning results on ImageNet using Inception v3 (21.9% top-1 error, 108MB) and MobileNet (29.4% top-1 error, 16.8 MB), and to our knowledge, we are the first to do so. Older CNNs such as AlexNet and VGGNet are heavily overparametarized and achieve lower accuracy compared to modern, efficient architectures like Inception and MobileNet, and we demonstrate the efficacy of pruning for model compression on compact, highly accurate state-of-the-art architectures. The fact that large, less accurate CNNs such as AlexNet or VGGNet can be pruned with minimal loss in accuracy does not directly imply that compact, highly accurate CNNs such as Inception or MobileNet can also be pruned with minimal loss in accuracy. We view the results obtained in this paper as significant since recent papers that attempt to prune a compact, highly accurate architecture trained on ImageNet achieve significantly worse results compared to us (Alvarez & Salzmann, 2017; and Dong et al., 2017 for the case of ResNet-50).

Han et al. (2015): Learning both Weights and Connections for Efficient Neural Networks. NIPS 2015.
Alvarez & Salzmann (2017): Compression-aware Training of Deep Networks. NIPS 2017.
Dong et al. (2017): Learning to Prune Deep Neural Networks via Layer-wise Optimal Brain Surgeon. NIPS 2017.

2. We propose a new gradual pruning technique that is simple and straightforward to apply across a variety of models/datasets with minimal tuning and can be seamlessly incorporated within the training process. Many existing pruning methods, like Han et al. (2015), depend on several stages of pruning and fine-tuning and require a lot of hyperparameter tuning, such as the per-layer weight threshold in each stage and the number of fine-tuning iterations. Using these pruning methods often requires a significant amount of ad-hoc tuning. In response to Reviewer 3’s request to see a comparison of our proposed gradual pruning method with the pruning method of Han et al. (2015), we cannot provide an exact comparison with Han et al.’s method because in Han et al.’s method, each layer’s weight threshold is chosen in an ad-hoc manner by specifying a “quality parameter” based on the "sensitivity of each layer to network pruning” (“how accuracy drops as parameters are pruned on a layer-by-layer basis”) and the sparsities of the weight tensors end up varying considerably between the different layers (16% to 91% for AlexNet and 42% to 96% for VGGNet) as a result of the per-layer "quality parameter". In our proposed method, we eliminate a lot of the hyperparameter tuning that is required in other pruning methods by embedding the pruning during the training process of the respective network which allows us to gradually prune and fine-tune according to one sparsity function across all of the layers (without requiring per-layer tuning). We demonstrate that our proposed gradual pruning technique works for several state-of-the-art CNNs and RNNs using the same sparsity function embedded in the training process of the respective network.

We have open-sourced the TensorFlow pruning library used to generate the results reported in this work. We believe this work makes an important contribution by showing that state-of-the-art neural network models can be pruned with minimal loss in accuracy using a new gradual pruning technique, incorporated as a part of the training procedure, that requires minimal tuning. We present compelling results showing that large-sparse models outperform small-dense models across several state-of-the-art neural network architectures (deep CNNs, stacked LSTMs, seq2seq models).

---

### Decision · Program_Chairs · 2018-01-29
**ICLR 2018 Conference Acceptance Decision**

**Decision:**

Invite to Workshop Track

**Comment:**

The authors present a thorough exploration of large-sparse models that are pruned down to a target size and show that these models can perform better than small dense models. Results are shown on a variety of datasets with as conv models and seq2seq. The authors even go so far as to release the code. I think the authors are to be thanked for their experimental contributions.
However, in terms of accepting the paper for a premier machine learning conference the method holds little surprise or non-obviousness. I think the paper is a good experimental contribution, and would make a good workshop paper instead but it offers little contribution by way of machine learning methods.